# *FutureDD*: Planning in POMDP with Encoded Future Dynamics

## Abstract

Partially observable Markov decision process (POMDP) is a powerful framework for modeling decision-making problems where agents do not have full access to environment states. In the realm of offline reinforcement learning (RL), agents need to extract policies on previously recorded decision-making datasets without directly interacting with environments. Due to the inherent partial observability of environments and the limited availability of offline data, agents must possess the capability to extract valuable insights from limited data, which can serve as crucial prior information for making informed decisions. Recent works have shown that deep generative models, particularly diffusion models, exhibit impressive performance in offline RL. However, most of these approaches mainly focus on fully observed environments while neglecting POMDPs, and heavily rely on history information for decision-making, disregarding the valuable prior information about the future that can be extracted from offline data. Having recognized this gap, we propose a novel framework *FutureDD* to extract future prior. *FutureDD* leverages an auxiliary prior model encoding future sub-trajectories to a latent variable, which serves as a compensation for directly modeling observations with a diffusion model. This enables *FutureDD* to extract richer prior information from limited offline data for agents to predict potential future dynamics. The experimental results on a set of tasks demonstrate that in the context of POMDPs, *FutureDD* provides a simple yet effective approach for agents to learn behaviours yielding higher returns.

## 1 Introduction

Planning under uncertainty is a crucial challenge for intelligent agents to accomplish real-world tasks due to various limitations such as noisy data collection and transmission. The partially observable Markov decision process (POMDP) stands out as a prominent framework for modeling such problems, with numerous works demonstrating its success in domains like robotics (Kurniawati, 2022; Pajarinen et al., 2022) and autonomous driving (Arbabi et al., 2023; Kuribayashi et al., 2023; Sunberg & Kochenderfer, 2022; Hubmann et al., 2017; Bai et al., 2015). However, a key challenge with POMDP environments is that when faced with incomplete state information, agents tend to find predicting potential future dynamics becomes increasingly stochastic. This underscores the pressing need for the agents to possess more prior information about the anticipated environmental shifts, which will enable them to make more informed decisions. Considering the offline reinforcement learning (RL) where agents extract policies from previously recorded data, both the inherent partial observability of environments and the limited availability of offline data pose an elevated challenge in obtaining a high-quality policy within the context of POMDPs.

The past few years have seen an increasing inclination towards formulating offline RL as a sequence prediction problem based on deep generative models such as autoregressive transformers (Katharopoulos et al., 2020; Vaswani et al., 2017) and diffusion models (Ho & Salimans, 2022; Dhariwal & Nichol, 2021). Numerous prior works (Chen et al., 2021; Janner et al., 2021; 2022; Zheng et al., 2022; Liu et al., 2022; Carroll et al., 2022; Ajay et al., 2022) have showcased that these sequence modeling approaches offer significant advantages over traditional TD-learning-based RL methods such as avoiding the issue of *deadly triad* (Van Hasselt et al., 2018) and can lead to improved performance outcomes, especially in fully observable environments. However, the POMDPs, crucial in many real-world applications, remain underexplored. Furthermore, directly transferring these outperforming-in-MDPs models to POMDPs may result in sub-optimal results (Ajay et al., 2022).

While most of these works are formulated as predicting future based on limited history information, agents planning in POMDPs demand richer prior information extracted from the offline data to analyze potential changes in the future.

In light of the above consideration, a natural idea to enrich the prior information is to extract potential future dynamics as a complement to limited history observations. This motivates us to propose *FutureDD*, a novel framework that delves into the exploration of leveraging future prior for sequential decision making in POMDPs with a deep generative model. The overall framework is in Figure 1. From an overall perspective, *FutureDD* consists of four core components: a prior model, a future encoder, a diffusion model and a MLP-based inverse dynamics model. The prior model is trained to extract potential future conditioned on the history information and return. The future encoder serves as a posterior model, which encodes the future sub-trajectories seen in the offline datasets during the training phase. The diffusion model learns the return-conditioned trajectory generation, where the trajectory is composed of observations and the corresponding future prior. The inverse model predicts actions based on history and the diffused future. During inference, we diffuse the trajectories with the prior future sampled from the prior model and take actions with the inverse model. In this manner, *FutureDD* equips agents with a future prior, enabling insightful and informed decision making in POMDPs where complete state information is unattainable.

We conduct a variety of experiments on Gym Mujoco tasks from the D4RL benchmark (Fu et al., 2020). Comparing *FutureDD* and its counterpart without future prior (Ajay et al., 2022), experimental results demonstrate that *FutureDD* exhibits outstanding performance on most tasks by introducing the future prior for POMDPs, particularly in the presence of diverse data qualities. We also compare *FutureDD* with a variant where the prior model is conditioned only on observations. The results show that the performance of *FutureDD* is more stable and consistently superior.

Overall, our main contributions can be summarized into three points. **(1)** We introduce *FutureDD* to address the sequential decision making problem in POMDPs, faced with challenges of the inherent partial observability and the limited offline data. **(2)** Motivated by extracting more prior information from the offline data as well as the neglected future information in datasets, we propose to leverage the future information as a prior for agents, which is realized by encoding the future sub-trajectories to a latent variable with the prior model and future encoder in *FutureDD*. **(3)** Experimental results on a set of Gym Mujoco tasks show that *FutureDD* has outstanding performance compared with other baselines in POMDP environments.

## 2 RELATED WORK

**Offline RL and Sequential Decision Making.** Offline reinforcement learning(RL) is a paradigm that extracts behavioral policies from a fixed dataset that was previously collected. Most work in this area focus on policy optimization, aiming to maximize the reward gained by performing the learnt policy. Motivated by the success of deep generative models in computer vision(Rombach et al., 2022; Saharia et al., 2022; Meng et al., 2021) and natural language processing (Gong et al., 2023; Brown et al., 2020; Devlin et al., 2019), there has been a growing trend to model offline RL as a sequence generation problem in the past few years. Chen et al. (2021) and Janner et al. (2021) concurrently explore to predict sequence based on autoregressive transformers, while the former focus on reward conditioning and the latter concentrate on beam-search-based planning. Zheng et al. (2022) propose a unified framework blending offline pretraining with online finetuning. Liu et al. (2022) employ masked autoencoder(MAE) to state-action trajectories and find randomly masking helps train a model generalizing well on several downstreams tasks. Apart from transformer-based methods, recent works Janner et al. (2022); Ajay et al. (2022); Dai et al. (2023) have also levaraged diffusion models, to aid planning in MDP environments.

**Diffusion Models for Decision Making.** Recent years, diffusion models have emerged as a prominent approach to generate high-quality samples with a wide range of applications, such as computer vision (Rombach et al., 2022; Saharia et al., 2022; Meng et al., 2021) and sequence modeling (Ajay et al., 2022; Janner et al., 2022). Training diffusion models involves a forward process transforming initial data distribution to a prior distribution and a reverse process that iteratively denoises the prior state back to the initial state using a neural network. To generate samples from a conditional distribution $p(x_0|c)$ given condition $c$, classifier guidance (Dhariwal & Nichol, 2021) and classifier-free guidance (Ho & Salimans, 2022) have been proposed to boost sample quality. In the field of sequence modeling,

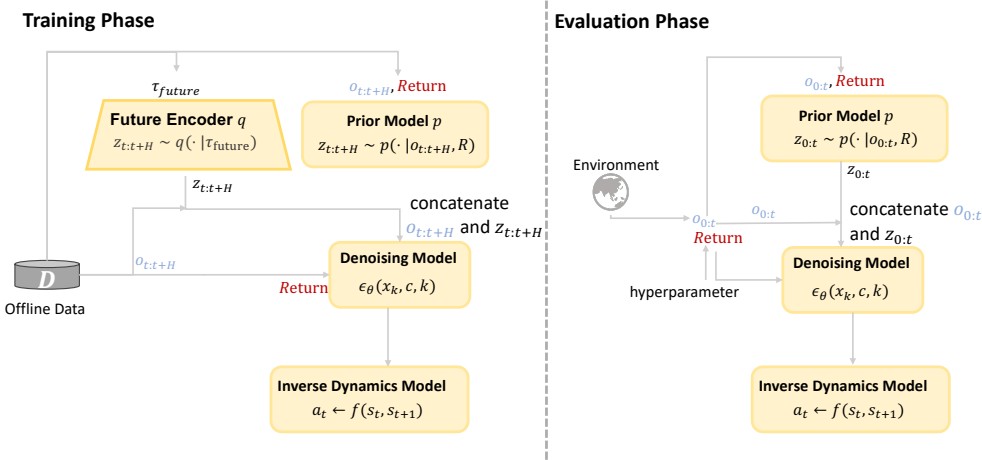

Figure 1: The overall framework of *FutureDD*. *FutureDD* consists of four core components: a prior model, a future encoder, a diffusion model and a MLP-based inverse dynamics model. During training, the future encoder $q$ encodes future sub-trajectories $\tau_{\text{future}}$ from the offline dataset $\mathcal{D}$ into latent variables $z$ as the future prior to be concatenated with the observations. The diffusion model learns the return-conditional distribution of concatenated observations and latent variables outputted by the future encoder. During evaluation, the prior model conditioned on observations and the target return, predicts the future prior $z$ to be utilized for planning. In this manner, *FutureDD* is enabled to be trained and evaluated to make decisions with future prior in POMDPs.

Janner et al. (2022) samples stacked states and actions under the guidance of a trained reward function while the work presented in Ajay et al. (2022) eliminate the need for training an additional reward function by utilizing a classifier-free guidance approach for state sampling, taking into account that states in RL tasks are typically continuous in nature, while actions exhibit more diversity. These works have indeed achieved impressive results in the context of MDP. However, they have somewhat overlooked the significance of POMDP environments. Given the remarkable performance demonstrated by Ajay et al. (2022) in MDP settings, our work builds upon their foundations.

**Encoding Future as Prior Information.** Leveraging future information is a popular and effective approach in Reinforcement Learning (RL). The utilized future information can encompass elements such as future returns or rewards (Chen et al., 2021; Schmidhuber, 2019; Kumar et al., 2019), goals (Liu et al., 2022; Andrychowicz et al., 2017), trajectory statistics (Furuta et al., 2021) and learned trajectory embeddings (Xie et al., 2023; Yang et al., 2022; Furuta et al., 2021). This information can be employed in various ways, including its use as a condition in future-conditioned supervised learning (Chen et al., 2021; Zheng et al., 2022) and future-conditioned unsupervised pretraining (Xie et al., 2023). Alternatively, it can be encoded as a latent variable to improve long-term prediction (Ke et al., 2019), address environmental stochasticity (Yang et al., 2022; Villaflor et al., 2022; Venuto et al., 2021), or aid function approximation for model-free RL (Venuto et al., 2021). Distinguished from prior works, our motivation lies in the extraction of valuable prior information from offline data to address the challenge of partial observability. And we consider the framework of modeling RL trajectories into conditional generation (Ajay et al., 2022) within the context of POMDP environments.

## 3 PROBLEM SETUP AND PRELIMINARIES

In this section, we firstly formulate the sequential decision making problem as a POMDP and give more detailed description of the definition. Secondly, we introduce the way to view decision making through the lens of conditional generative modeling (Ajay et al., 2022). The above parts are setup for our proposed *FutureDD* in Section 4.

### 3.1 Sequential decision making in POMDPs

We formulate the sequential decision making problem as a discounted partially observable Markov decision process defined by the tuple $(\mathcal{S}, \mathcal{O}, \mathcal{A}, \mathcal{T}, \mathcal{R}, \mathcal{Z}, \gamma)$, where $\mathcal{S}, \mathcal{O}, \mathcal{A}$ denote the state, observation and action spaces respectively. The transition function, $\mathcal{T} : \mathcal{S} \times \mathcal{A} \to \mathcal{S}$, defines the transition between states when an action is taken. The reward function $\mathcal{R} : \mathcal{S} \times \mathcal{A} \times \mathcal{S} \to \mathbb{R}$ specifies the reward when a transition happens. $\mathcal{Z} : \mathcal{S} \to \mathcal{O}$ denotes the observation emission model describing the probability distribution that maps the unseen state to its corresponding observation. $\gamma$ is the discount factor (Puterman, 2014). The RL objective of a agent in POMDP is to find the policy maximizing the expected return of a trajectory $\tau := \{(o_t, a_t, r_t)\}_{t=0}^{T}$, which could be formulated as:

$$\pi^* = \arg\max \mathbb{E}_{a_t \sim \pi, \; s_{t+1} \sim \mathcal{T}(\cdot|s_t,a_t), r_t \sim \mathcal{R}(s_t,a_t,s_{t+1})} \left[ \Sigma_{t=0}^{T} \gamma^t r_t \right]. \tag{1}$$

### 3.2 Diffusion models

To learn the data distribution $q(\boldsymbol{x})$, a diffusion model (Sohl-Dickstein et al., 2015; Ho et al., 2020) consists of two Markov chains: a predefined forward process $q(\boldsymbol{x}_k|\boldsymbol{x}_{k-1}) := \mathcal{N}\left(\boldsymbol{x}_k; \sqrt{1-\beta_k}\boldsymbol{x}_{k-1}, \beta_t\boldsymbol{I}\right)$ perturbing data to noise, and a reverse denoising process learning transition kernels parameterized by trainable neural networks, which take the form of $p(\boldsymbol{x}_{k-1}|\boldsymbol{x}_k) := \mathcal{N}(\boldsymbol{x}_{k-1}; \mu_\theta(\boldsymbol{x}_k, k), \Sigma_\theta(\boldsymbol{x}_k, k))$. The variance schedule $\beta_k \in (0, 1)$ is a carefully chosen hyperparameter, and $\mathcal{N}(\mu, \Sigma)$ denotes a Gaussian distribution with mean $\mu$ and variance $\Sigma$. The prior distribution is defined as $p(\boldsymbol{x}_K) := \mathcal{N}(\boldsymbol{x}_K; \mathbf{0}, \boldsymbol{I})$ with a long enough $K$. Diffusion models can be trained by the mean square loss between the predicted noise $\epsilon_\theta(\boldsymbol{x}_k, k)$ and the noise $\epsilon$ sampled from $\mathcal{N}(0, \boldsymbol{I})$ proposed by Ho et al. (2020). For conditional sample generation with classifier-free guidance (Ho & Salimans, 2022), the predicted noise of diffusion models can be extended to both an unconditional $\epsilon_\theta(\boldsymbol{x}_k, \emptyset, k)$ and a conditional $\epsilon_\theta(\boldsymbol{x}_k, \boldsymbol{c}, k)$, where $\emptyset$ represents a dummy value. Diffusion models sample conditional data using the perturb noise

$$\epsilon_\theta(\boldsymbol{x}_k, k) + \omega(\epsilon_\theta(\boldsymbol{x}_k, \boldsymbol{c}, k) - \epsilon_\theta(\boldsymbol{x}_k, \emptyset, k)), \tag{2}$$

in which $\omega$ describes the guidance strength given condition $\boldsymbol{c}$. In this work, we apply the conditional diffusion models to model the states of a trajectory conditioned on returns $\boldsymbol{c} := R(\tau)$, following the approach by Ajay et al. (2022), which is defined as $\boldsymbol{x}_k(\tau) := (o_t, o_{t+1}, ..., o_{t+H-1})_k$ with $k$ denoting the noising step and $t$ denoting the timestep in trajectory $\tau$ of horizon $H$. During the training phase, the return $R(\tau)$ is defined as the discounted accumulated reward of $\tau$ while during evaluation return $R$ is a predefined hyperparameter.

## 4 Methodology

In this section, we present *FutureDD*, which learns policies with future prior from offline data in POMDP environments. We begin by providing an overview of the *FutureDD* framework, followed by detailed explanations of its individual components. After that, we give a comprehensive exposition of the training and inference processes of *FutureDD*.

### 4.1 *FutureDD*: An Overview and Component Breakdown

**Overview of *FutureDD*.** The overall framework is shown in Figure 1. To achieve of the goal of planning with future prior, we first need a prior model capable of extracting future information based on history information. However, the challenge lies in how to provide appropriate future information to train the prior model. Both the availability and quality of future information are crucial for effectively training the whole framework. Considering the availability, we can utilize the future sub-trajectories recoreded in offline data. Motivated by prior work (Xie et al., 2023), we leverage a future encoder to encode the actual future sub-trajectories from offline data into latent space. This encoded future serves as the prior information to train the prior model. As for the quality of future information, it's essential to condition not only on history but also on return. This dual conditioning ensures that the model has access to comprehensive information for more accurate and robust learning. Apart from the prior model and the future encoder, we utilize a diffusion model to predict the future and an inverse dynamics model to predict actions based on the observations and the future prior.

---

**Algorithm 1:** Planning with *FutureDD* in POMDPs

---

1 **Input:** Prior model $p_{\theta_1}$, noise model $\epsilon_{\theta_3}$, inverse dynamics model $f_{\theta_4}$, guidance intensity $\omega$, noise scale $\alpha$, history length $c$, condition return $R$, denoising steps $K$

2 Initialize $\tau_{\text{past}} \leftarrow$ `Queue(length=c)`;

3 $t \leftarrow 0$;

4 **while** *not done* **do**

5     Get observation $o_t$;

6     $z_t \leftarrow p_{\theta_1}(o_t, R)$;                  `// get the future prior z`

7     $s_t \leftarrow [o_t, z_t]$;

8     $\tau_{\text{past}}$.`insert`$(s_t)$;

9     Initialize $\boldsymbol{x}_K(\tau) \sim \mathcal{N}(0, \alpha\boldsymbol{I})$;

10     **for** $k = K, K-1, ..., 1$ **do**      `// Diffuse with observation and prior`

11         $\boldsymbol{x}_k(\tau)[:\text{length}(\tau_{\text{past}})] \leftarrow \tau_{\text{past}}$;

12         $\hat{\epsilon} \leftarrow \epsilon_{\theta_3}(\boldsymbol{x}_k(\tau), \emptyset, k) + \omega(\epsilon_{\theta_3}(\boldsymbol{x}_k(\tau), \emptyset, k) - \epsilon_{\theta_3}(\boldsymbol{x}_k(\tau), R, k))$;

13         $\mu_{k-1}, \Sigma_{k-1} \leftarrow$ `Denoise`$(\boldsymbol{x}_k(\tau), \hat{\epsilon})$;

14         $\boldsymbol{x}_k(\tau) \sim \mathcal{N}(\mu_{k-1}, \alpha\Sigma_{k-1})$;

15     Extract $(s_t, s_{t+1})$ from $\boldsymbol{x}_0(\tau)$;

16     Take action $a_t \leftarrow f_{\theta_4}(s_t, s_{t+1})$;    `// Predict action with inverse model`

17     $t \leftarrow t+1$;

---

Overall, our proposed *FutureDD* comprises four main components, which can be classified into two major categories. The first category includes a future encoder and prior model, which are used for providing future information. The second category consists of the diffusion model and MLP-based inverse dynamics model, which are utilized for planning based on both historical and extracted future information.

**The Prior Model and Future Encoder.** We jointly train the future encoder compressing the future sub-trajectories into the latent space and the prior model extracting the future prior. And the prior model is trained to predict the future prior $z$ conditioned on observations and returns, which can be formulated as $z \sim p(\cdot|o, R)$ with $p(\cdot|o, R)$ outputting a multivariate Gaussian distribution.

From an episode $\tau$ of length $T$ in the offline dataset, a sub-trajectory $\tau_{t:t+H}$ of length $H$ is sampled as

$$\tau_{t:t+H} := (o_t, a_t, r_t, o_{t+1}, a_{t+1}, r_{t+1}, ..., o_{t+H-1}, a_{t+H-1}, r_{t+H-1}). \tag{3}$$

The return of $\tau_{t:t+H}$ can be represented as $R(\tau_{t:t+H})$. And the corresponding future sub-trajectory to be embedded into the latent space can be denoted as

$$\tau_{\text{future}} := (o_{t+u}, a_{t+u}, o_{t+u+1}, a_{t+u+1}, ..., o_{t+u+H-1}, a_{t+u+H-1}), \tag{4}$$

where $u$ is an integer denoting the distance between the history horizon and the future horizon. However, we only embed the future dynamics consisting of the observations and actions. With the prior model and future encoder, the training objective of predicting the future prior is to minimize

$$\mathcal{L}_{\text{future}} := \beta\mathbb{E}_{\tau\sim\mathcal{D}, z\sim q_\theta(\cdot|\tau_{\text{future}})}[D_{\text{KL}}(q_\theta(\cdot \mid \tau_{\text{future}})\|\mathcal{N}(\boldsymbol{0}, I))] \tag{5}$$

$$+ \mathbb{E}_{\tau\sim\mathcal{D}, z\sim q_\theta(\cdot|\tau_{\text{future}})}[D_{\text{KL}}(\lfloor q_\theta(z \mid \tau_{\text{future}})\rfloor\|p_\theta(z \mid o_t, R(\tau)))], \tag{6}$$

inspired by previous work (Xie et al., 2023). Here, $\mathcal{D}_{\text{KL}}$ denotes the Kullback-Leibler divergence and $\lfloor\cdot\rfloor$ describes the stop-gradient operator. The former term of the future loss is a regularization term to control the capacity of $z$ and to avoid that the future encoder fails to capture the full distribution of future (Xie et al., 2023; Higgins et al., 2016), where $\beta$ is a carefully chosen hyperparameter as the regularization coefficient. The second term is for training the prior model to extract the future prior conditioned on history observations and returns.

**The Diffusion Model and Inverse Dynamics Model.** The diffusion model is trained to forecast the subsequent observation from history observations. However, if we directly apply the diffuser to the observations $o_{t:t+H}$ in the dataset as $\boldsymbol{x}_k(\tau) := (o_t, o_{t+1}, ..., o_{t+H-1})_k$ representing incomplete environmental states typically, it may lead to suboptimal outcomes (Ajay et al., 2022). This

underscores the significant function of the future prior prior as a complement to the part of unseen states for agents. Therefore, instead of directly modeling the raw observations in the offline dataset, *FutureDD* utilizes a concatenation of the observations and the corresponding latent variables $z$. For simplification, we denote the augmented observations as $s_t := (o_t, z_t)$, and the target data to be modelled by the diffusion model as $\boldsymbol{x}_k(\tau) := (s_t, s_{t+1}, ..., s_{t+H-1})_k$. The simplified training objective of the diffusion model (Ho et al., 2020) can be represented as

$$\mathcal{L}_{\text{diff}} := \mathbb{E}_{\epsilon \sim \mathcal{N}(0, \boldsymbol{I})} \left[ \| \epsilon - \epsilon_\theta(\boldsymbol{x}_k, \boldsymbol{c}, k) \|^2 \right] \tag{7}$$

where condtition $\boldsymbol{c}$ can be empty or return $R(\tau)$.

At each timestep, *FutureDD* extracts $(s_t, s_{t+1})$ from $\boldsymbol{x}_0(\tau)$ after $K$ denoising steps, and a MLP-based inverse dynamics model $f(s_t, s_{t+1})$ is utilized to take both of them as input and predict the current action $\hat{a}_t := f(s_t, s_{t+1})$ with the training objective of

$$\mathcal{L}_{\text{inv}} := \| a_t - f(s_t, s_{t+1}) \|^2. \tag{8}$$

## 4.2 TRAINING AND EVALUATION

**Training.** To train *FutureDD* with four components, we need to consider all the losses mentioned above. Having sub-trajectories and the corresponding future sub-trajectories from the offline dataset, the latent variables $z$ can be sampled from the output distributions from the future encoder and be used for training the prior model. After concatenating observations with the latent variables, the combination is leveraged for training the diffusion model and the inverse dynamics model. The overall training objective is

$$\mathcal{L}_{FutureDD} := \mathcal{L}_{\text{future}} + \lambda_1 \cdot \mathcal{L}_{\text{diff}} + \lambda_2 \cdot \mathcal{L}_{\text{inv}} \tag{9}$$

where $\lambda_1$ and $\lambda_2$ are weights of $\mathcal{L}_{\text{diff}}$ and $\mathcal{L}_{\text{inv}}$ respectively.

**Evaluation.** During evaluation, the future piror is sampled from the learned prior model and is concatenated with observations to predict the future in the same way as the training process. The overall pipeline of planning with *FutureDD* is outlined in Algorithm 1.

## 5 EXPERIMENTS

In this section, we evaluate the performance of *FutureDD*. Our primary aim is to explore two key questions: 1) Can providing more future prior information assist the agent in learning a superior policy from offline data? 2) Does a reward conditioned prior model demonstrate better performance compared to a prior model without reward conditioning?

**Baselines.** To answer the first question, we evaluate the performance of *FutureDD* compared with *Decision Diffuser* (Ajay et al., 2022) which models observations with a diffusion model. Since the diffusion model and inverse model of *FutureDD* are built on *Decision Diffuser* (*DD*), the difference between *FutureDD* and *DD* lies in whether the future information is leveraged. Apart from that, we also design a variant of *FutureDD*, the *FutureDD_w/oR* with the prior model that is conditioned on observations solely.

**Implementation Details.** We use an autoregressive transformer as the future encoder mapping future sub-trajecotries to a set of latent variables. And the prior model we use in *FutureDD* is a single layer neural network mapping the embedded observations and returns to the future prior. The denoising model and inverse dynamics model are built on *Decision Diffuser*. The former model $\epsilon_\theta$ is parameterized by a temporal U-net composed of a set of convolutional residual blocks. The final loss weights $\lambda_1$ and $\lambda_2$ we adopt are set to 1.

**Datasets and Environments.** The experiments are conducted in 3 Gym Mujoco environments with a total of 9 datasets from the D4RL benchmark (Fu et al., 2020). Each environment includes datasets of `"medium replay"`, `"medium"` and `"medium expert"`. The `"medium replay"` datasets consist of all the observed samples in the replay buffer during training until the polices reach "medium" level, while the `"medium"` datasets are all generated by "medium" level policies. The `"medium expert"` datasets are composed of optimal and suboptimal data. All the observations to be leveraged come from the original states excluded with 2 dims. In this manner, we investigate the performance of *FutureDD* in POMDP environments.

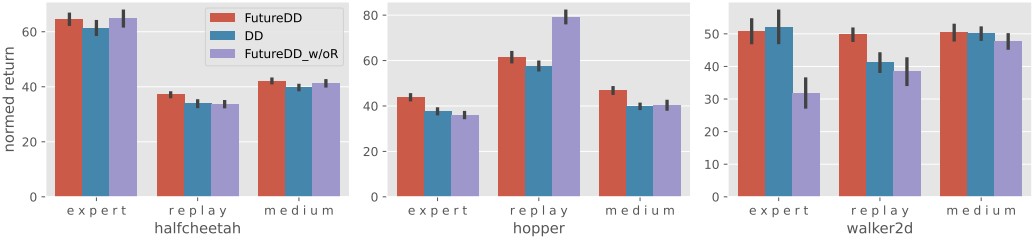

Figure 2: **Experimental results on 9 datasets**. These results are the average and variance of evaluations under 10 random seeds with each evaluation consisting of 10 episodes. Among the 9 datasets, *FutureDD* outperforms other baselines in **7/9** environments.

**Main Results and Analysis.** The results are in Figure 2. While *FutureDD* outperforms *DD* on **8/9** environments, *FutureDD* performs the best in **7/9** environments. First, with *FutureDD* outperforming *DD* on **8/9** environments, it reveals that leveraging additional future information can improve the policy learned from offline data in POMDPs. However, the degree of this improvement may vary with the quality of datasets. It is noted that on the three `"medium expert"` datasets, the improvement from prior information is relatively marginal. This observations are attributed to the `"medium expert"` datasets themselves inherently containing optimal or sub-optimal information. Extra prior information may not substantially alter the quality of policy, and could potentially even be detrimental. This is evident from the performance of *FutureDD_w/oR* on the three `"medium expert"` datasets, where it exhibits a notable decline in two out of three datasets compared to *DD*. In this scenario, the prior information inadvertently introduces bias during model training, harming the performance of the policy. However, when comparing *FutureDD* with *FutureDD_w/oR*, incorporating return in the condition helps mitigate the adverse impact of bias when prior information is not as essential for policy learning. This adjustment ultimately solidifies *FutureDD* as the superior performer across all nine datasets, ensuring its robustness against the negative influences of additional bias.

Therefore, a reasonable and efficient prior model can significantly improve the performance and robustness of the algorithm, while inappropriate prior information might introduce unnecessary bias, negatively affecting the learning process and the final performance. Overall, the experimental results prove that FutureDD can effectively utilize prior information to learn better policies across various environments.

## 6 CONCLUSION

In this paper, we introduce *FutureDD*, a novel framework that extracts future prior for conditional generative modeling of decision making in POMDP environments. Our experiments conducted acorss diverse datasets have empirically validated the efficacy of *FutureDD*. The additional future prior information has been shown to sustantially aid agents in learning return-maxmizing policies in POMDP environments, demonstrating the effectiveness of *FutureDD*. Moreover, the reward conditioning of the prior model further bolsters the performance of the prior model, as evidenced by the consistent outperformance of *FutureDD* compared with *FutureDD_w/oR*. Our findings substantiate the critical role of utilizing the future information to extract prior for decision making in POMDPs where complete state information is out of reach.

**Limitation.** Despite the promising results in our work, there are also some limitations in *FutureDD*. The focus of *FutureDD* is to leverage the future prior for decision making in POMDPs and predict actions with a simple MDP-based inverse dynamics model following the work by Ajay et al. (2022). However, some previous works (Paischer et al., 2022) show that memorizing history information is also crucial for POMDPs. Therefore, the inverse dynamics model utilized in our work may lead to constraints of the capacity of the whole framwork. Replacing MLP with a model capable of simultaneously utilizing both longer history information and rich future prior should be an interesting direction to explore.

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
