# OpenReview forum: "FutureDD: Planning in POMDP with Encoded Future Dynamics"
_ICLR.cc/2024/Conference — ICLR 2024 Conference Withdrawn Submission_

### Official Review · Reviewer_e8st · 2023-10-23

**Soundness:** 2 fair
**Presentation:** 1 poor
**Contribution:** 1 poor
**Rating:** 3
**Confidence:** 2

**Summary:**

This work extends the decision diffuser (DD) and Pretrained Decision Transformer (PDT) into the POMDP setting. FutureDD has two main components. The first component includes the prior model and future encoder, which are jointly optimized to capture future information. The second component includes diffusion model and inverse dynamics model. Together FutureDD is proposed and evaluated on D4RL 3 environments (9 datasets) against DD.

**Strengths:**

The FutureDD is novel and original in that it is conditioned on future information compared to DD.

**Weaknesses:**

I did not understand the methodology part of this work, partly because I am not familiar with decision diffuser, but also because the writing is not clear to me. I encourage the authors to write the formulation and objectives more clearly (e.g., add input parameters to every loss function, define every model with its input and output, provide some visual interaction with these models), which is also helpful for readers to understand its novelty.

I think the inverse dynamics loss is problematic in POMDPs. First, we cannot observe s_t in POMDPs, but it is written in the framework and methodology. Second, s_t seems a function of o_t in algorithm 1, which is not sufficient to predict a_t given o_t, o_{t+1}. We need to use history.

The content of this work is far from expectation for an ICLR paper. It has only 7 pages without an appendix. As it is a methodology work, it needs to be evaluated on more domains (see the DD paper). The results are also not significantly better as the paper claimed. Figure 2 shows that Future DD is very close to other baselines (also should add PDT).

Some types (condtition, piror, framwork) should be fixed.

**Questions:**

1. I don’t get the future loss objective in Equation 5 and 6. For example, why should we separate tau_{t:t+H} and tau_future? What’s the range of u? The objective seems very similar to Equation 2 in the PDT paper, what’s the novelty and why should we change some term?
2. The diffusion objective in Equation 7 and 8 seems very similar to DD paper, again what’s the novelty?
3. Which 2 dims are excluded in D4RL?

---

### Official Review · Reviewer_BsjN · 2023-10-30

**Soundness:** 3 good
**Presentation:** 2 fair
**Contribution:** 2 fair
**Rating:** 3
**Confidence:** 5

**Summary:**

This study extends the Decision Diffuser (DD) method to incorporate future information into the trajectory, aiming to enhance performance in partially observable environments. The proposed approach employs a diffusion model during training to encode information about future trajectory steps. This diffusion model learns an inference distribution over latent variables capable of encoding future information, which is subsequently utilized in training the DD. During evaluation, a prior distribution is learned to sample these latent variables. Experimental evaluation was conducted using D4RL MuJoCo environments, and the results were compared against the DD baseline.

**Strengths:**

- This work presents a straightforward and user-friendly extension to the DD method.
- The methodology is robust, building upon a well-established approach known for enhancing performance in partially observable environments.
- The method is explained well and put into context with other related works.

**Weaknesses:**

- The experimental evaluation is poor, showing only marginal improvements over DD.
- The study focuses on a limited number of MuJoCo tasks, primarily the simplest ones, which are all continuous control tasks.  None of these tasks have partial observability as the main challenge.
- In terms of novelty, this work appears somewhat limited when compared to similar future encoding methods like PGIF (Venuto et al., 2021) or Furuta et al. (2021). The only distinction seems to be its application within the context of Decision Diffuser.

I recommend the rejection of this work due to its limited novelty, inadequate experimental evaluation, and the lack of significant improvements.

**Questions:**

Can the authors expand the number of environments in experiments and highlight further improvements?  Is it possible to focus on environments where partial observability is the main challenge?

---

### Official Review · Reviewer_bCks · 2023-11-03

**Soundness:** 3 good
**Presentation:** 3 good
**Contribution:** 2 fair
**Rating:** 5
**Confidence:** 4

**Summary:**

This paper proposes an offline learning framework for decision making. The proposed FutureDD contains four parts: future encoder, prior model, diffusion model and an inverse dynamic model. The key idea is to leverage future dynamic information to enrich current state information. Experiments demonstrate that incorporating future information as prior indeed improves the policy performance.

**Strengths:**

1. This paper propose a new approach to leverage information about future dynamics to enrich current state information.
2. The paper is well orginized and easy to follow.

**Weaknesses:**

Despite the merits of this paper, I have several concerns on soundness and novelty on the proposed method. Moreover, the experiments are not adequate to justify the effectiveness and necessity of each part of the model.

1. The paper aims to solve the planning problem in POMDPs, but I did not see any special design for the partial observability. It is always beneficial to incorporate future information to the current state, even when the current state is fully observable. Therefore, I think it is not proper to claim that the proposed method targets the POMDP problem, unless your method compensates some state information that should have been observed in full observability.

2. The proposed FutureDD framework actually contains four sub-models, which bring more trick issues during implementation. I think that the authors should clearly explain the necessity and advantage of using each of the sub-models. There are some works that also incorporate future information to enrich state information. For example, [1] also uses the future reward signal. What are the advantages of FutureDD compared to these existing methods which also use future information?

3. Diffusion models are usually time-consuming in decision tasks. What is the ratio of time cost of the diffusion model to the whole decision process? It seems that the diffusion model in this paper is only used to predict $s_{t+1}$. Can the diffusion model be replaced by another simpler model?

4. The experiments only demonstrate that incorporating future information is beneficial, which has already been shown in many existing works. It is crucial to show that FutureDD performs better than other offline learning methods that also use future information, such as [1] and Decision Transformers.



[1] RETURN-BASED CONTRASTIVE REPRESENTATION LEARNING FOR REINFORCEMENT LEARNING, ICLR 2021
LEARNING FOR REINFORCEMENT LEARNING

**Questions:**

1. How do you set the distance between the history horizon and the future horizon (u) during training?
2. How do you set the target reward R during evaluation?

---

### Official Review · Reviewer_sLyp · 2023-11-05

**Soundness:** 3 good
**Presentation:** 2 fair
**Contribution:** 2 fair
**Rating:** 5
**Confidence:** 4

**Summary:**

This paper proposes a new framework called FutureDD for decision making in partially observable Markov decision processes (POMDPs) in the context of offline reinforcement learning. The key idea is that learning an explicit latent representation of future states that depends on past observations and a specified return condition can provide additional information about future. This is done by closely following the approach proposed in [1], which leverages sub-trajectories from offline dataset during training to (i) learn a future encoder that encodes actual future sub trajectories (of observation, action pairs) into a latent space, and (ii) learn a prior model that can generate a future embedding based on the observations to date and a given return condition. This future prior then serves as additional input to condition a diffusion model, in conjunction with an inverse dynamics pipeline, similar to the approach in Decision Diffusion [2] to predict next actions. The authors perform experiments on three different datasets on each one of the 3 Gym Mujoco environments in D4RL. They use a subset of states as observations to demonstrate the improved performance of FutureDD on POMDP problems.

[1] Zhihui Xie, Zichuan Lin, Deheng Ye, Qiang Fu, Yang Wei, and Shuai Li. Future-conditioned unsupervised pretraining for decision transformer. In International Conference on Machine Learning, pp. 38187–38203. PMLR, 2023.

[2] Anurag Ajay, Yilun Du, Abhi Gupta, Joshua Tenenbaum, Tommi Jaakkola, and Pulkit Agrawal. Is conditional generative modeling all you need for decision-making? arXiv preprint arXiv:2211.15657, 2022.

**Strengths:**

- The idea of utilizing encoding of future sub-trajectories as prior information with decision diffuser, particularly in the context of partially observable environments seems novel.

**Weaknesses:**

- I am not convinced that the experiments show that FutureDD provides significant improvements over DD.
    - Although the authors claim that FutureDD outperforms the Decision Diffusion (DD) in 8 out of 9 environments, the performance gains appear to be statistically significant in only 4 of these cases.
    - Further, it is not completely clear which two 2 dimensions where excluded from the MuJoCo environments to make it a POMDP - I believe the choice of the excluded dimensions can have significant impact on the results. Were the two dimensions selected randomly?
    - It is not clear if the authors tuned the hyper-parameters of the Decision Diffusion (DD) baseline for the new partially observable environments. In my experience, the impact of this is non-trivial and can significantly change the presented results

- It’s not completely clear how future encoder and prior model are designed. If I understand it correctly, future encoder encodes a sub-trajectory of observation action pairs, and the prior model is expected to retrieve appropriate latent variables based on observation history and a return condition
    - Why is the prior model conditioned on the return to retrieve latent variables that encode a sub-trajectory without reward values? Intuitively, the prior model shouldn’t be conditioned on the return, right?
    - The results suggest that the prior model, even without reward conditioning, significantly outperforms both FutureDD and DD in one of the experiments. Were any further experiments performed to understand which factors determine if reward conditioning prior model is relevant?
    - Which specific return—whether it be from the history sub-trajectory, the future sub-trajectory, or the entire trajectory—is the prior model conditioned upon during training?
    - How accurate were the latent variable predictions from the prior model?

**Questions:**

In addition to the questions in the weaknesses section, I have the following questions/suggestions:

- I am curious how FutureDD compares with DD in fully observable environments, i.e., MDPs. Were any experiments performed in this direction? If so, was there any observed performance improvement?
- Have the authors experimented with different future encoder architectures in addition to the autoregressive transformer? Was there any hyperparameter search?
- Loss weighting hyperparameters like λ1 and λ2  seem to be set without much discussion of tuning. Was there any sensitivity analysis? What was the impact?
- Could the authors provide a more detailed analysis of the dynamics that the future latent encoding captures? Are there other ways to evaluate or visualize this?
- I believe that Figure 1 should be improved. It was difficult to understand the illustration at first glance without reading through the paper, which somewhat defeats the purpose of having an illustrative figure.

---

### Author Response · Authors · 2023-11-22

We sincerely thank all the reviewers for the valuable feedback and we think additional time is needed to refine our work. So we decide to withdraw the paper after careful consideration.